

# Prognostic significance and tumor-immune infiltration of mTOR in clear cell renal cell carcinoma

Na Li[1], Jie Chen[1,2], Qiang Liu[3], Hongyi Qu[4], Xiaoqing Yang[5], Peng Gao[3], Yao Wang[3], Huayu Gao[4], Hong Wang[1] and Zuohui Zhao[4]

[1] Department of Urology, The First Affiliated Hospital of Shandong First Medical University & Shandong Provincial Qianfoshan Hospital, Jinan, Shandong, China
[2] Department of Urology, Jinan Central Hospital, Cheeloo College of Medicine, Shandong University, Jinan, Shandong, China
[3] Department of Cardiology, The First Affiliated Hospital of Shandong First Medical University & Shandong Provincial Qianfoshan Hospital, Jinan, Shandong, China
[4] Department of Pediatric Surgery, The First Affiliated Hospital of Shandong First Medical University & Shandong Provincial Qianfoshan Hospital, Shandong Engineering and Technology Research Center for Pediatric Drug Development, Jinan, Shandong, China
[5] Department of Pathology, The First Affiliated Hospital of Shandong First Medical University & Shandong Provincial Qianfoshan Hospital, Jinan, Shandong, China

Corresponding authors
Hong Wang,
wanghong@sdhospital.com.cn
Zuohui Zhao, zhaozuohui@126.com

## ABSTRACT

Mammalian target of rapamycin (mTOR), a serine/threonine kinase involved in cell proliferation, survival, metabolism and immunity, was reportedly activated in various cancers. However, the clinical role of mTOR in renal cell carcinoma (RCC) is controversial. Here we detected the expression and prognosis of total mTOR and phosphorylated mTOR (p-mTOR) in clear cell RCC (ccRCC) patients, and explored the interactions between mTOR and immune infiltrates in ccRCC. The protein level of mTOR and p-mTOR was determined by western blotting (WB), and their expression was evaluated in 145 ccRCC and 13 non-tumor specimens by immunohistochemistry (IHC). The relationship to immune infiltration of mTOR was further investigated using TIMER and TISIDB databases, respectively. WB demonstrated the ratio of p-mTOR to mTOR was higher in ccRCC than adjacent specimens ($n = 3$), and IHC analysis elucidated that p-mTOR expression was positively correlated with tumor size, stage and metastasis status, and negatively correlated with cancer-specific survival (CSS). In univariate analysis, high grade, large tumor, advanced stage, metastasis, and high p-mTOR expression were recognized as prognostic factors of poorer CSS, and multivariate survival analysis elucidated that tumor stage, p-mTOR and metastasis were of prognostic value for CSS in ccRCC patients. Further TIMER and TISIDB analyses uncovered that mTOR gene expression was significantly associated with numerous immune cells and immunoinhibitors in patients with ccRCC. Collectively, these findings revealed p-mTOR was identified as an independent predictor of poor survival, and mTOR was associated with tumor immune infiltrates in ccRCC patients, which validated mTOR could be implicated in the initiation and progression of ccRCC.

## INTRODUCTION

Cancer has become the first or second leading cause of death and one major burden worldwide nowadays (*Siegel et al., 2021*; *Sung et al., 2021*). Kidney cancer is a common urological neoplasm, with estimated 431,288 newly diagnosed cases and 179,368 cancer-related deaths worldwide in 2020 (*Sung et al., 2021*). Renal cell carcinoma (RCC), the most common malignancy within the kidney, comprises a heterogeneous group of tumors including clear cell (70~80%), papillary (10~15%), chromophobe (3~5%), and other less common subtypes (*Rini, Campbell & Escudier, 2009*). Radical nephrectomy is the gold standard for localized RCC, while systemic therapy is the main course of treatment for patients with advanced RCC, especially metastatic RCC (mRCC) (*Posadas, Limvorasak & Figlin, 2017*). As RCC is resistant to traditional chemotherapy and radiotherapy, medical therapy for mRCC has transitioned from a nonspecific immune approach (cytokine therapy) to novel molecular targeted therapy, including tyrosine kinase inhibitors (TKIs), mammalian target of rapamycin (mTOR) inhibitors, and immune checkpoint inhibitors (*Barata & Rini, 2017*). Multiple TKIs, especially Sunitinib, are widely used pharmacologic agents for mRCC patients (*Posadas, Limvorasak & Figlin, 2017*). Two mTOR inhibitors, Temsirolimus as a first-line therapy and Everolimus as a second-line therapy, have been approved for clinical use in mRCC patients. Immune checkpoint inhibitors such as programmed cell death protein 1 (PD-1) inhibitor, Nivolumab and programmed death-ligand 1(PD-L1) inhibitor Avelumab, directly reverse the adaptive tumor cell deploy to avoid host immunity, thus attenuate immune response and postpone cancer progression (*Makhov et al., 2018*; *Posadas, Limvorasak & Figlin, 2017*). However, molecular targeted therapy for RCC remains limited due to varied response rate and apparent adverse effects, such as fatigue, diarrhoea and hyperglycaemia (*Posadas, Limvorasak & Figlin, 2017*). Of particular note, the mechanisms underlying the different responsiveness and side-effect might be complicated. Combined therapy with multiple molecular-targeted drugs has attracted much attention (*Posadas, Limvorasak & Figlin, 2017*). Thus, there is an urgent requirement to further investigate the molecular mechanisms that drive RCC initiation and progression, which would facilitate to realize individual or personalized therapy for this heterogeneous tumor.

mTOR, a highly conserved serine/threonine kinase, is a core component of phosphatidylinositol three kinase (PI3K)/AKT/mTOR signaling pathway. mTOR encompasses two functionally distinct protein complexes: mTOR complex 1 (mTORC1) and mTORC2. mTORC1 contains rapamycin-sensitive RAPTOR (regulatory associated protein of mTOR), while mTORC2 contains rapamycin-insensitive companion of mTOR (RICTOR). mTOR regulates diverse biological processes, including cell proliferation, survival, metabolism, and immunity under physiological and pathological conditions (*Posadas, Limvorasak & Figlin, 2017*; *Saxton & Sabatini, 2017*). mTOR is generally activated (characterized by phosphorylation at Ser2,448) to catalyze the phosphorylation

of its downstream targets and modulate their activity, such as ribosomal S6 protein kinase 1 (S6K1), eukaryotic translation initiation factor 4E binding protein 1(4E-BP1), and protein kinase C (PKC), thereby regulating protein synthesis, cell growth, and metabolism (*Posadas, Limvorasak & Figlin, 2017*). Recent researches also focus on its role in cancer. mTOR was reported to be aberrantly activated during tumorigenesis, and it was found to have played a crucial role in the initiation and progression of various malignant tumors, including breast, prostate, lung, liver, and kidney cancers (*Posadas, Limvorasak & Figlin, 2017*). Due to its characterized activity, mTOR has been recognized as a target for cancer therapy.

The advent of high throughput technologies and massive public databases, such as The Cancer Proteome Atlas (TCPA), The Cancer Genome Atlas (TCGA), and some online cancer genome tools, such as Tumor Immune Estimation Resource (TIMER), Tumor-Immune System Interactions and DrugBank (TISIDB), have facilitated to elaborate the complex molecular mechanism in cancer (*Li et al., 2013*; *Li et al., 2017*; *Ru et al., 2019*). Previous studies demonstrated that phosphorylated mTOR (p-mTOR) was increased in RCC, and enhanced p-mTOR was associated with impaired overall survival (OS), which indicated that the PI3K/AKT/mTOR signaling pathway could promote the initiation and progression of RCC (*Darwish et al., 2013*; *Kruck et al., 2010*; *Liontos et al., 2017*; *Rausch et al., 2019*). Recent large-scale proteogenomic analyses found that mTOR and/or p-mTOR expression levels were significantly correlated with improved outcomes in RCC (*Fan et al., 2020*; *Zhang et al., 2017*), which was contradictory to the most widely accepted research and data. Collectively, the above studies failed to reach a consistent conclusion, and the prognostic role of mTOR in RCC remains controversial. In the current study, we detected total mTOR and p-mTOR expression in three pairs clear cell RCC (ccRCC) and their corresponding non-tumor kidney specimens using western blotting (WB), examined mTOR (including p-mTOR) expression in 145 ccRCC and 13 adjacent nonneoplastic kidney specimens using immunohistochemistry (IHC) with tissue microarray (TMA), explored the relationship between mTOR (including p-mTOR) expression and the clinicopathological parameters (including patients' age and tumor size), and elucidated the complex tumor-immune interactions in ccRCC using TIMER and TISIDB databases, which tried to investigate the clinical role of mTOR and p-mTOR in ccRCC patients.

## MATERIALS & METHODS

### Patients

The research was approved by the ethical committees of The First Affiliated Hospital of Shandong First Medical University (2017-S007), and all the participants signed the informed consents. A total of 148 ccRCC patients with resectable tumors between March 2010 and January 2015 were enrolled (*Yuan et al., 2020*). All the ccRCC samples were primary lesions, and they were verified using hematoxylin and eosin (HE) staining by two pathologists (X.Q. Yang and Y. Wang) after surgery. The cohort #1 was used to compare mTOR and p-mTOR expression by WB analysis, which consisted of three cases of ccRCC and their corresponding normal kidney specimens (one female and two males, age

**Table 1 Correlation between mTOR (including p-mTOR) expression and clinicopathological parameters of ccRCC (n = 145).**

| Parameters | p-mTOR staining | | $\chi^2$ | P-value | mTOR staining | | $\chi^2$ | P-value |
|---|---|---|---|---|---|---|---|---|
| | Low, n (%) | High, n (%) | | | Low, n (%) | High, n (%) | | |
| Sex | | | | | | | | |
| Male (n = 109) | 49 (44.95) | 60 (55.05) | | | 55 (50.346) | 54 (49.54) | | |
| Female (n = 36) | 23 (63.89) | 13 (36.11) | 3.881 | 0.056 | 18 (50.00) | 18 (50.00) | 0.002 | 0.962 |
| Age | | | | | | | | |
| <60 yrs (n = 65) | 36 (46.15) | 29 (53.85) | | | 32 (49.23) | 33 (50.77) | | |
| ≥60 yrs (n = 80) | 36 (45.00) | 44 (55.00) | 1.547 | 0.244 | 41 (51.25) | 39 (48.75) | 0.058 | 0.868 |
| ISUP grade | | | | | | | | |
| G 1 (n = 26) | 16 (61.54) | 10 (38.46) | | | 11 (42.31) | 15 (57.69) | | |
| G 2 (n = 56) | 27 (48.21) | 29 (51.79) | | | 27 (48.21) | 29 (51.79) | | |
| G 3–4 (n = 63) | 29 (46.03) | 34 (53.97) | 1.846 | 0.397 | 35 (55.56) | 28 (44.44) | 1.458 | 0.482 |
| AJCC stage | | | | | | | | |
| T I (n = 42) | 33 (78.57) | 9 (21.43) | | | 22 (52.38) | 20 (47.62) | | |
| T II (n = 38) | 22 (57.89) | 16 (42.11) | | | 15 (39.47) | 23 (60.53) | | |
| T III–IV (n = 65) | 17 (26.15) | 48 (73.85) | 29.441 | <0.001 | 36 (55.38) | 29 (44.62) | 2.527 | 0.283 |
| Tumor size | | | | | | | | |
| <7.0 cm (n = 57) | 38 (66.67) | 19 (33.33) | | | 30 (52.63) | 27 (47.37) | | |
| ≥7.0 cm (n = 88) | 34 (38.64) | 54 (61.36) | 10.872 | 0.001 | 43 (48.86) | 45 (51.14) | 0.196 | 0.735 |
| Metastasis | | | | | | | | |
| Negative (n = 121) | 67 (55.37) | 54 (44.63) | | | 60 (49.59) | 61 (50.41) | | |
| Positive (n = 24) | 5 (20.83) | 19 (79.17) | 9.557 | 0.003 | 13 (54.17) | 11 (45.83) | 0.168 | 0.824 |
| Survival rate | | | | | | | | |
| Alive (n = 51) | 41 (80.39) | 10 (19.61) | | | 22 (43.14) | 29 (56.86) | | |
| Dead (n = 94) | 31 (32.98) | 63 (67.02) | 29.731 | <0.001 | 51 (54.26) | 43 (45.74) | 1.635 | 0.201 |

Notes:
Statistical analyses were performed using Pearson chi-Square tests.
Abbreviation: ISUP, international society of urological pathology; AJCC, American Joint Committee on Cancer.

from 55 to 65, International Society of Urological Pathology (ISUP) grading with 1 G1 + 1 G2 + 1 G3, American Joint Committee on Cancer (AJCC) staging with 1 TI + 2 TII). The cohort #2 was used to evaluate the prognostic value of mTOR and p-mTOR by IHC assay. TMA from 145 ccRCC and 13 non-tumor specimens was created as described previously (*Yuan et al., 2020*). The patient characteristics (such as tumor grade, stage, patients' sex and age) and follow-up period were recorded (Table 1). The median period of follow-up was 47.2 months, ranging from 20 to 76 months. Among the HE-stained slides, 10 archived paraffin-embedded tissue sections (including four females and six males, age from 45 to 68, ISUP grading with 2 G1 + 5 G2 + 3 G3, AJCC staging with 3 TI + 6 TII +1 TIII) were randomly selected and used to examine the immune cells.

## WB analysis

Three pairs ccRCC and adjacent tissues were chosen for WB analysis (*Yuan et al., 2020*). Membranes were incubated with the desired primary antibodies: mTOR (1:2,000, rabbit,

ab32028; Abcam, Cambridge, MA, USA), p-mTOR (Ser2448, 1:2,000, rabbit, ab109268; Abcam, Cambridge, MA, USA), anti-Actin (1:1,000, mouse, BM0627; BOSTER, Beijing, China), the corresponding secondary antibodies (SA00001-2 & SA00001-1; Proteintech, Wuhan, China) were diluted to 1:5,000, and enhanced chemiluminescence (Amersham Imager 600; Marlborough, MA, USA) was used for immunodetection as previously described (*Yuan et al., 2020*).

### IHC analysis

TMA was constructed from 145 ccRCC and 13 adjacent kidney specimens, which was used for IHC analysis as previously described (*Zhao et al., 2016*). The slides were stained with the described primary antibodies: mTOR (1:100) and p-mTOR (1:100), and diaminobenzidine (ZLI-9017; Zhongshan, Beijing, China) was used for visualization. Human placenta served as negative controls (*Rausch et al., 2019*).

The staining was analyzed by two independent reviewers who were unaware of the disease outcome. The expression level of mTOR and p-mTOR was evaluated and scored according to their staining intensity (0~3) and frequency (0~4) as previously described (*Yuan et al., 2020*). Accordingly, they were categorized into 2 groups: high group (staining scores ≥3 for p-mTOR, ≥2 for mTOR) and low group (scores <3 for p-mTOR, <2 for mTOR).

### TIMER database analysis

TIMER database (https://cistrome.shinyapps.io/timer) was used to explore the association between mTOR gene expression and immune cell infiltrates of ccRCC, which estimated the abundance of tumor-infiltrating immune cells (TIICs) from TCGA, as previously described (*Li et al., 2017*). In brief, *Gene* module was chosen to analyze the complex tumor–immune interactions between mTOR and six TIIC subsets, *i.e.*, B cells, CD4 + T cells, CD8+ T cells, macrophages, neutrophils, and dendritic cells, in the ccRCC cohort ($n$ = 533). purity adjusted was selected, and the scatter plots of Spearman's correlations between them were displayed.

### TISIDB database analysis

TISIDB database (http://cis.hku.hk/TISIDB), which integrated multiple public databases including TCGA, was also used to reveal the immune infiltration of mTOR in cancer, as previously described (*Ru et al., 2019*). In this study, *Lymphocyte* and *Immunomodulator* modules were selected to investigated the correlations between mTOR gene expression and abundance of tumor-infiltrating lymphocytes (TILs) & immunoinhibitors in ccRCC ($n$ = 534), then the heatmaps and scatter plots between mTOR and 28 TILs & 24 immunoinhibitors from Charoentong's study, were displayed subsequently.

### Statistical analysis

All analysis was carried out using SPSS 21.0 software (SPSS Inc., Chicago, IL, USA). For WB analysis, the differential expression of mTOR and p-mTOR was compared by means of Student's t-test. Correlations between mTOR (including p-mTOR) expression and clinical parameters were calculated using Pearson Chi-square test. The survival

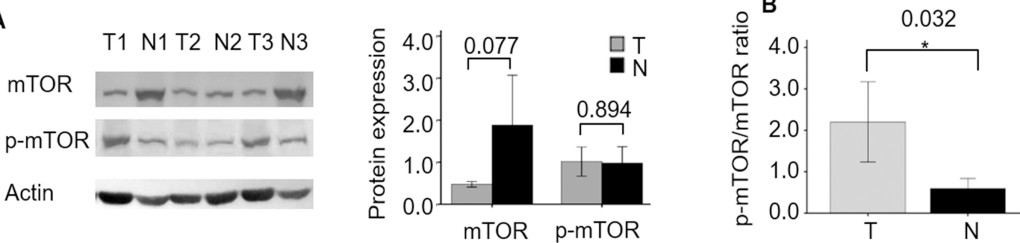

**Figure 1 mTOR and p-mTOR expression between ccRCC and adjacent normal kidney tissues by WB analysis.** (A) Relative expression levels of mTOR and p-mTOR in three pairs ccRCC (T) and their adjacent normal kidney (N) tissues. Left: WB bands; right: statistical bar chart. (B) The ratio of p-mTOR to mTOR in ccRCC (T) and normal kidney (N) tissues. $^*P < 0.05$.

curves for cancer-specific survival (CSS) were calculated using Kaplan–Meier method, and prognostic factors were analyzed using Cox proportional hazard regression model. For TIMER and TISIDB analyses, Spearman's correlations analysis was used to infer the association between mTOR and tumor immune infiltrates. $P < 0.05$ was considered statistically significant.

## RESULTS

### mTOR and p-mTOR expression in ccRCC

First, we compared the protein expression of mTOR and p-mTOR between three cases of ccRCC and their adjacent non-tumor specimens. WB demonstrated that the expression level of mTOR showed a decreasing trend in ccRCC compared with adjacent specimens ($P = 0.077$, Fig. 1), which was consistent with its mRNA expression (*Fan et al., 2020*). While p-mTOR expression was stable in both malignant and adjacent non-tumor tissues ($P = 0.894$). When normalized to mTOR expression, we found the ratio of p-mTOR to mTOR elevated 4.259 folds in cancerous than adjacent kidney specimens ($P = 0.032$), which validated previous report (*Kruck et al., 2010*).

### The relationship between mTOR (including p-mTOR) expression and clinicopathological parameters of ccRCC

Then we analyzed mTOR (including p-mTOR) expression and its association with patient characteristics (Table 1). IHC analysis on TMA showed that mTOR positive signal was distributed in majority of the adjacent non-tumor kidney tubule epithelial cells and sparse malignant cells, and p-mTOR positive signal was aggregately scattered in nonneoplastic kidney tubule epithelial cells and cancer cells (Fig. 2). Relative weak cytoplasm staining for mTOR and strong staining for p-mTOR expression were seen in the malignant cells of kidney. To be specific, stronger positive staining with p-mTOR was examined in 73 (50.34%) cases and weaker staining was examined in 72 (49.66%) cases of ccRCC tissues, respectively. p-mTOR expression was higher in large tumors (≥7.0 cm) than small ones (<7.0 cm), the difference was statistically significant ($P = 0.001$). Moreover, enhanced p-mTOR expression (≥3 scores) was positively correlated with advanced stage (stage III–IV, $P < 0.001$) and metastatic status (M1, $P = 0.003$), and it was negatively

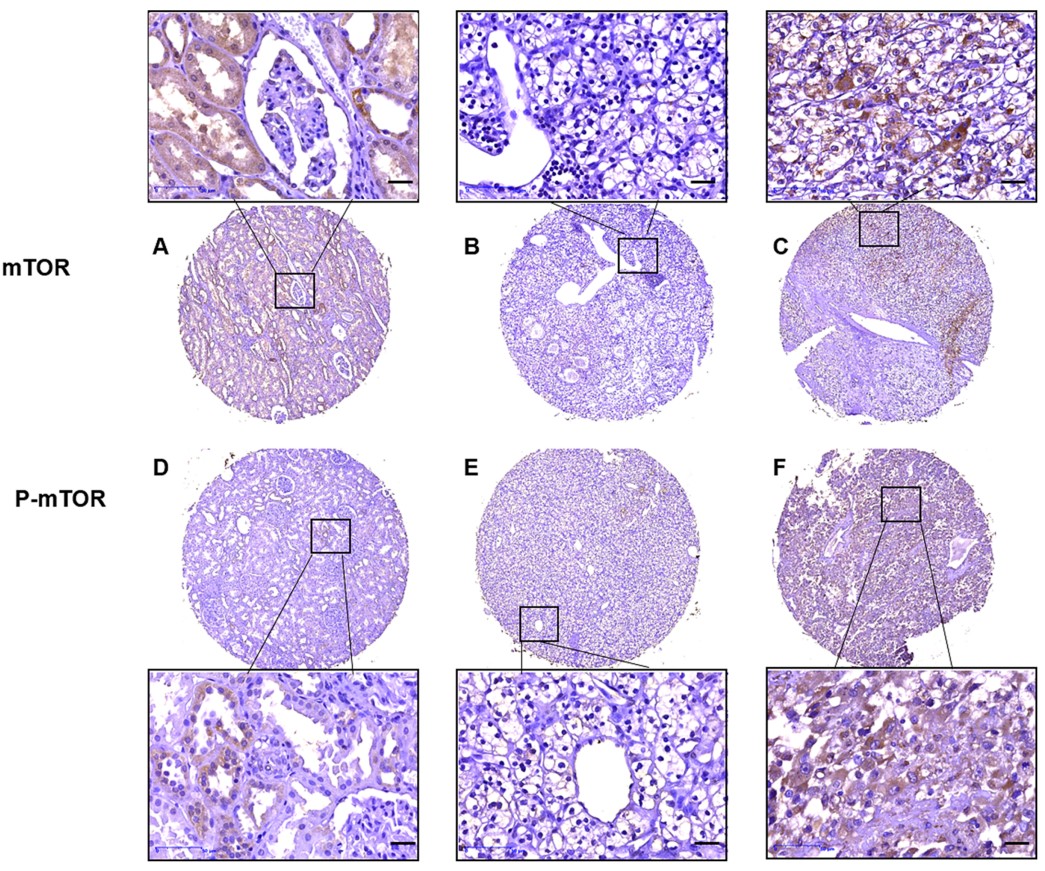

**Figure 2 Representative immunostaining photomicrographs of mTOR and p-mTOR expression by IHC analysis.** Staining signals displayed cytoplasmic localization of mTOR in adjacent kidney ((A) staining score 6) and ccRCC tissues ((B) low expression, score 0, (C) high expression, score 6), and p-mTOR in adjacent kidney ((D), score 4) and ccRCC tissues ((E) low expression, score 3, (F) high expression, score 7). The corresponding magnified areas in ccRCC and adjacent tissues also showed in the top and bottom, respectively. Original magnification 200×; bars, 50 μm.

correlated with CSS rate ($P < 0.001$). The relationship between p-mTOR expression and patients' sex, age or tumor grade was not significant ($P = 0.056$, $P = 0.244$, $P = 0.397$, respectively). As for mTOR, its enhanced expression (≥2 scores) was not significantly correlated with any clinicopathological parameters such as tumor grade, stage, size, metastasis, survival rate, patients' sex or age ($P > 0.05$), which was also observed in previous study (*Rausch et al., 2019*). To sum up, the above data elucidated that p-mTOR expression was positively correlated with tumor size, pathological stage and metastasis status, and inversely correlated with CSS, which indicated that p-mTOR could be a predictor of tumor aggressiveness for ccRCC patients (*Rausch et al., 2019*).

## p-mTOR, but not mTOR, was a poor prognostic factor of CSS in ccRCC patients

During ~4 years follow-up, Kaplan–Meier survival curve displayed that CSS rate was higher in patients with low p-mTOR expression (score < 3) than those with high

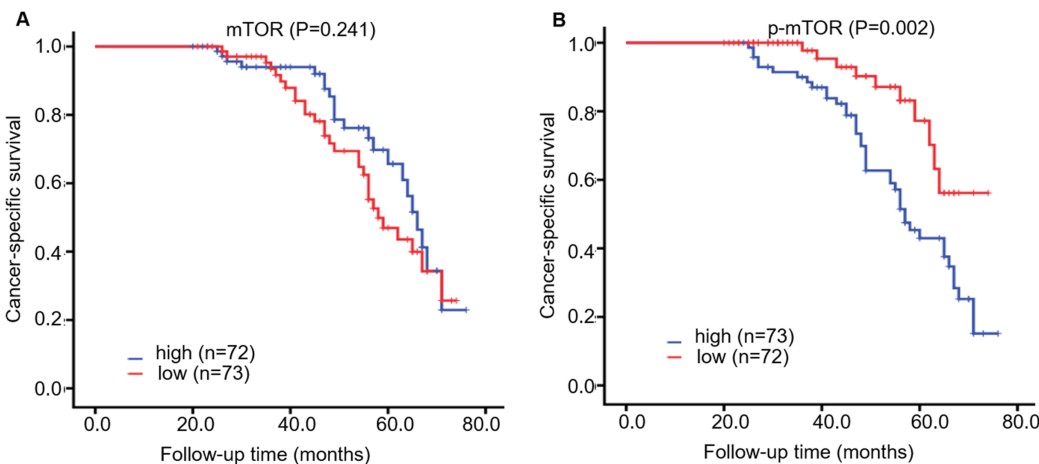

**Figure 3 Kaplan–Meier survival curves demonstrated cancer-specific survival of 145 patients with ccRCC, according to mTOR (A) and p-mTOR (B) staining.** Blue and red curves represented high and low staining, respectively.

**Table 2 Univariate and multivariate survival analysis of cancer-specific survival ($n$ = 145).**

| Parameters | Univariate[a] HR (95% CI)[b] | P-value | Multivariate[a] HR (95% CI)[b] | P-value |
|---|---|---|---|---|
| Sex | 1.305 [0.834–2.042] | 0.245 | 1.452 [0.892–2.364] | 0.134 |
| Age | 1.511 [0.990–2.306] | 0.055 | 1.438 [0.908–2.279] | 0.122 |
| Grade (G3–4)[c] | 2.152 [1.573–2.943] | <0.001 | 2.187 [1.526–3.133] | <0.001 |
| Stage (TIII–IV)[d] | 4.869 [3.285–7.217] | <0.001 | 3.812 [2.460–5.907] | <0.001 |
| Size (≥7.0 cm) | 2.414 [1.492–3.906] | <0.001 | 1.216 [0.713–2.073] | 0.472 |
| Metastasis | 9.060 [5.179–15.849] | <0.001 | 2.894 [1.588–5.282] | 0.001 |
| High p-mTOR | 2.933 [1.891–4.549] | <0.001 | 1.733 [1.037–2.897] | 0.036 |
| High mTOR | 0.882 [0.586–1.327] | 0.547 | 1.365 [0.876–2.128] | 0.170 |

**Notes:**
[a] Statistical analysis by Cox proportional hazards regression model.
[b] Abbreviation: HR, hazard ratio; CI, confidence interval.
[c] For grade: 1, 2 vs 3–4.
[d] For stage: I, II vs III–IV.

expression (≥3) (log-rank = 10.008, $P$ = 0.002), while CSS rate was not correlated with mTOR expression in 145 ccRCC patients (log-rank = 1.373, $P$ = 0.241, Fig. 3). Then we further investigated the role of mTOR (including p-mTOR) on tumor prognosis using Cox regression analysis. Univariate survival analysis demonstrated that high p-mTOR expression was associated with a shorter CSS in patients with ccRCC (hazard ratio (HR) 2.933, 95% confidence interval (CI) [1.891–4.549], $P$ < 0.001, Table 2). In the meantime, it manifested that advanced stage (HR 4.869, 95% CI [3.285–7.217], $P$ < 0.001), high grade (HR 2.152, 95% CI [1.573–2.943], $P$ < 0.001), large tumor (HR 2.414, 95% CI [1.492–3.906], $P$ < 0.001), metastasis (HR 9.060, CI [5.179–15.849], $P$ < 0.001) were all correlated with poor prognosis for CSS. As for mTOR, its expression was not associated with CSS ($P$ = 0.547). And CSS was not associated with patients' sex or age ($P$ = 0.245, $P$ = 0.055, respectively).

 

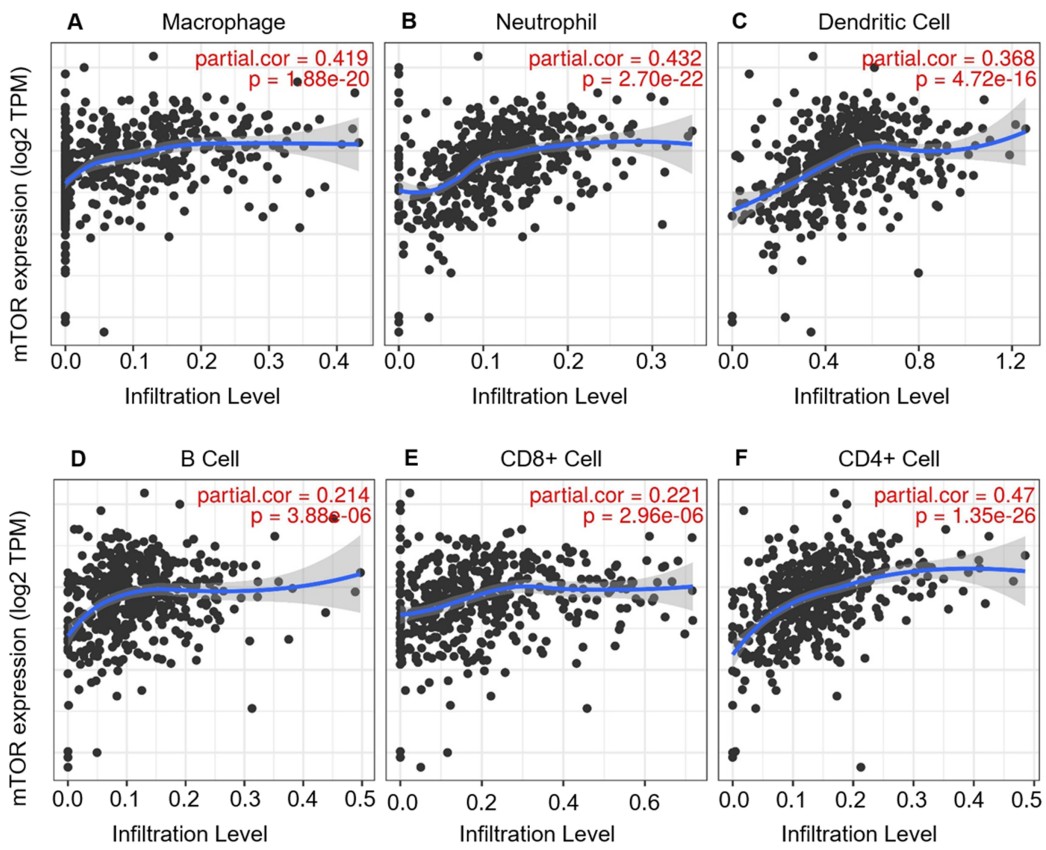

**Figure 4 Correlation between mTOR expression and tumor-infiltrating immune cells (TIICs) in 533 ccRCC patients (TIMER).** The infiltration levels of the six TIIC subsets, *i.e.*, macrophage (A), neutrophil (B), dendritic cell (C), B cell (D), CD8+ T cell (E) and CD4+ T cell (F). TPM: transcripts per million.

Further multivariate survival analysis displayed that tumor stage (HR 3.812, 95% CI [2.460–5.907], *P* < 0.001), grade (HR 2.187, 95% CI [1.526–3.133], *P* < 0.001), metastasis (HR 2.894, 95% CI [1.585–5.282], *P* = 0.001) and p-mTOR (HR 1.733, 95% CI [1.037–2.897], *P* = 0.036), were all recognized as independent predictors for CSS in patients with ccRCC (Table 2), while tumor size, mTOR expression, patients' age, or sex were not identified as prognostic factors (*P* = 0.134, *P* = 0.122, *P* = 0.472, *P* = 0.170, respectively, Table 2).

## TIMER and TISIDB analyses revealed the relationship between mTOR and immune infiltrates in ccRCC

ccRCC is an immunotherapy-sensitive tumor, and mTOR regulates tumor immunity (*Posadas, Limvorasak & Figlin, 2017*; *Saxton & Sabatini, 2017*). On the base of the expression and prognosis of mTOR in ccRCC, we further evaluated the correlation between mTOR and immune features, such as immune cells and immunomodulators, in ccRCC using TIMER and TISIDB databases. TIMER analysis showed that mTOR expression was positively correlated with infiltrating levels of all the six TIIC subsets, *i.e.*,

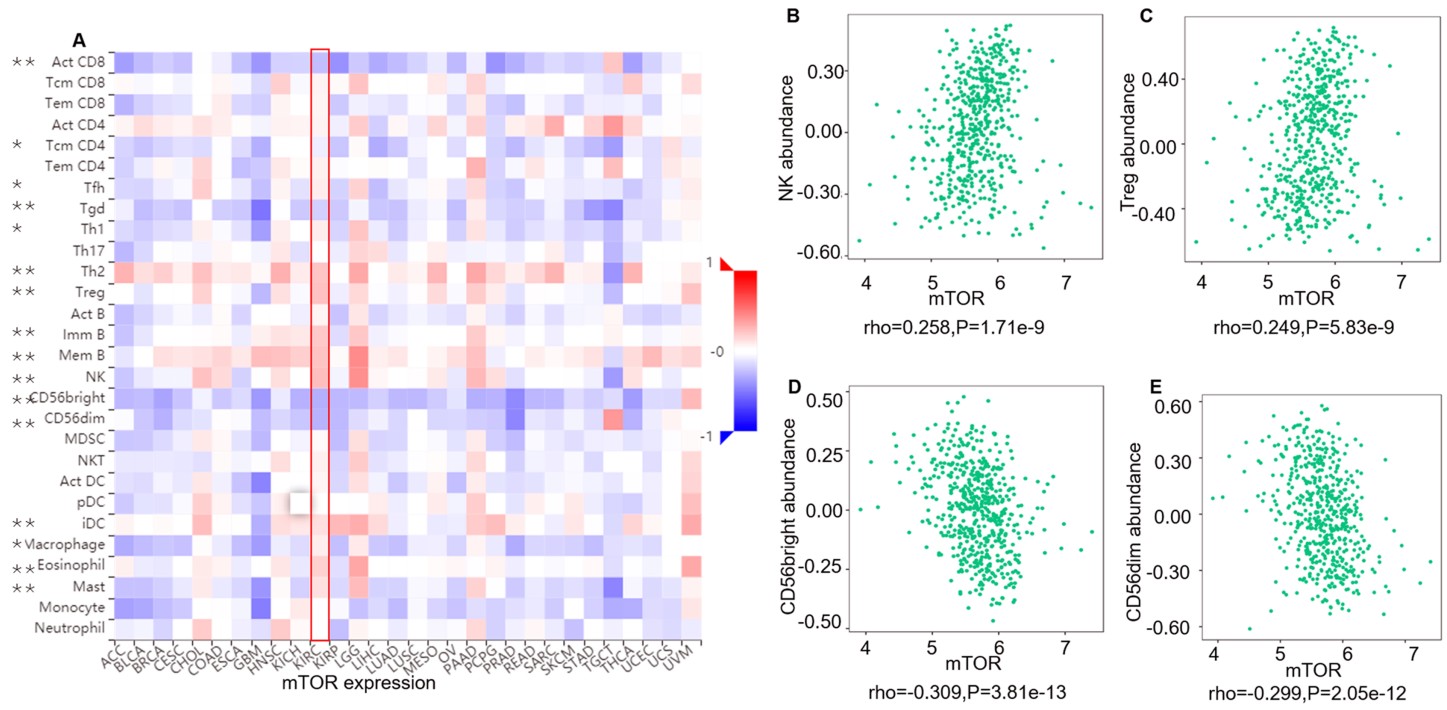

**Figure 5 Correlation between mTOR expression and lymphocytes in 534 ccRCC patients (TISIDB).** (A) The pan-cancer analysis of relationship between mTOR expression and abundance of the 28 tumor-infiltrating lymphocytes (TILs). The top four TILs either positive ((B) NK cell, (C) Treg cell) or negative ((D) CD56bright cell; (E) CD56dim cell) correlation with mTOR expression in ccRCC patients. $^*P < 0.05$, $^{**}P < 0.01$.

macrophage (rho = 0.419, $P < 0.001$), neutrophil (rho = 0.432, $P < 0.001$), dendritic cell (rho = 0.368, $P < 0.001$), B cell (rho = 0.214, $P < 0.001$), CD8+ T cell (rho = 0.221, $P < 0.001$), and CD4+ T cell (rho = 0.470, $P < 0.001$) among the 533 ccRCC cases (Fig. 4). At the same time, TISIDB analysis demonstrated that mTOR expression was significantly correlated to the abundance of numerous TILs in 534 ccRCC cases (Fig. 5). Specifically, mTOR expression was positively related with the abundance of natural killer cell (NK, rho = 0.258, $P < 0.001$), regulatory T cell (Treg, rho = 0.249, $P < 0.001$), and was negatively correlated with the abundance of CD56 bright natural killer cell (CD56bright, rho = −0.309, $P < 0.001$) and CD56 dim natural killer cell (CD56dim, rho = −0.299, $P < 0.001$). Then we evaluated the association between mTOR expression and the abundance of 24 immunoinhibitors across human cancers, which was illustrated in Fig. 6. Particularly, the immunoinhibitors displaying the greatest positively correlations included CD274 (PD-L1, rho = 0.364, $P < 0.001$), transforming growth factor-β receptor type I (TGFBR1, rho = 0.290, $P < 0.001$), colony stimulating factor-1 receptor (CSF1R, rho = 0.254, $P < 0.001$), while PDCD1 (PD-1) was not significantly associated with mTOR in ccRCC (rho = −0.064, $P = 0.137$). HE staining demonstrated that the immune cells were sporadically scattered in the cancerous tissues, especially in the tissues with higher p-mTOR expression (Fig. S1), which confirmed the rich immune infiltrates in the ccRCC tissues (*Diaz-Montero, Rini & Finke, 2020*). The above results implied that mTOR could be

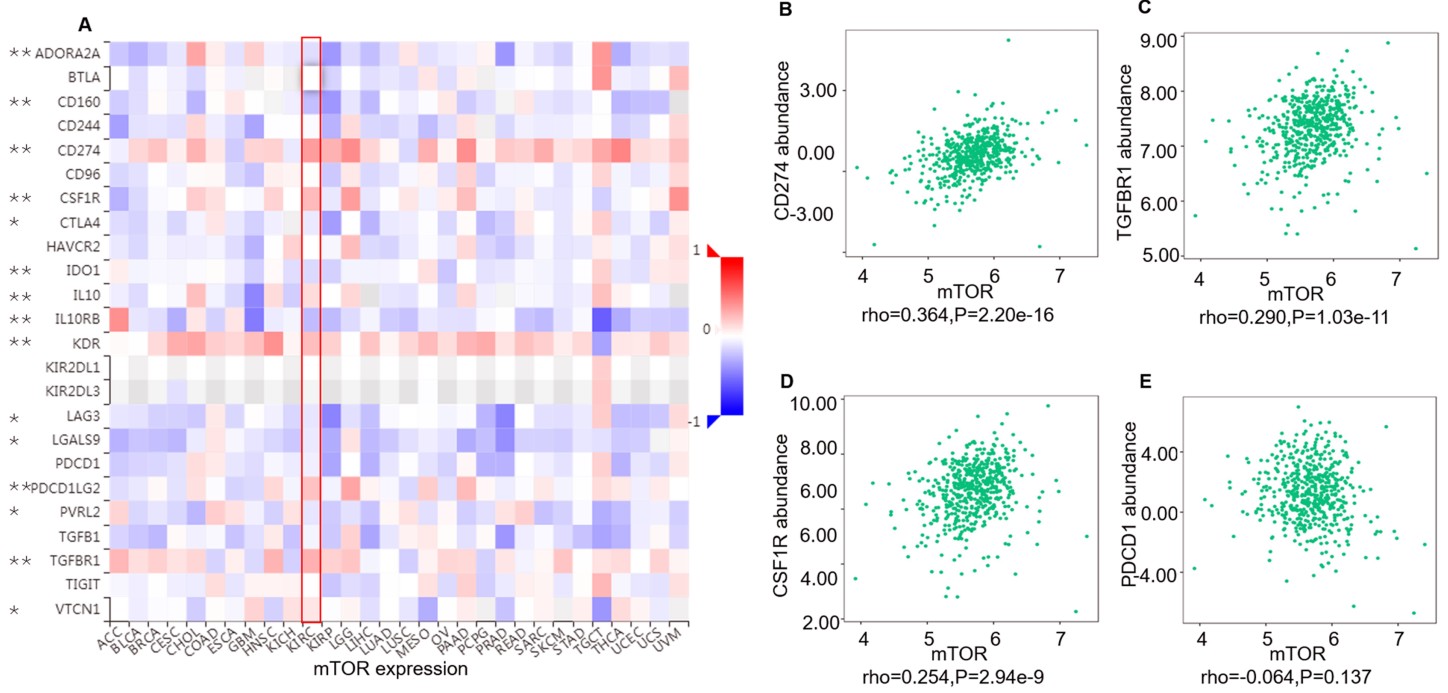

**Figure 6 Correlation between mTOR expression and immunoinhibitors in 534 ccRCC patients (TISIDB).** (A) The pan-cancer analysis of relationship between mTOR expression and abundance of the 24 immunoinhibitors. The top three immunoinhibitors ((B) CD274/PD-L1, (C) TGFBR1, (D) CSF1R) and PDCD1 ((E), PD-1) correlation with mTOR expression in ccRCC patients. *$P < 0.05$, **$P < 0.01$.

involved in regulating the immune infiltrates in ccRCC patients, which was consistent with previous reports (*Diaz-Montero, Rini & Finke, 2020*; *Zhang et al., 2019*).

## DISCUSSION

PI3K/AKT/mTOR pathway plays a pivotal role in cancer pathogenesis and progression, and mTOR is frequently activated in various malignancies, such as breast, prostate, lung, liver, and kidney cancers (*Posadas, Limvorasak & Figlin, 2017*). Increasing literature has elaborated mTOR activation in RCC, while its clinical value during RCC pathogenesis has not been fully elucidated yet (*Cancer Genome Atlas Research Network, 2013*; *Chen et al., 2016*; *Darwish et al., 2013*; *Han et al., 2017*; *Zhang et al., 2017*). Our previous study revealed that expression of mTOR pathway members might play distinguished roles in different stages or grades of RCC, implying the complexity of mTOR signaling pathway in the development of RCC (*Fan et al., 2020*). Thus, it is necessary to present a better looking of the relationship between mTOR signaling pathway and RCC. In the current study, we compared the expression level of mTOR and p-mTOR between ccRCC and adjacent kidney tissues, evaluated its prognostic significance using TMA, then revealed the tumor-immune interaction of mTOR in ccRCC using online databases (TIMER and TISIDB) and HE staining. Our study validated mTOR activation in ccRCC, *i.e.*, the ratio of p-mTOR to mTOR was increased in ccRCC ($P < 0.05$), which was also observed in previous report (*Kruck et al., 2010*). Subsequently IHC analysis demonstrated that p-mTOR expression was positively correlated with tumor size, pathological stage, and

metastasis status ($P = 0.001$, $P < 0.001$, $P = 0.003$, respectively), and negatively correlated with CSS ($P < 0.001$), which implied elevated p-mTOR was significantly correlated with the aggressiveness of ccRCC (*Rausch et al., 2019*). Then survival analysis elucidated that p-mTOR was identified as an independent predictor of poor CSS in 145 patients with ccRCC ($P < 0.05$). Further TIMER and TISIDB databases analysis revealed mTOR could regulate numerous immune cells (including TIICs and TILs) and immunoinhibitors in >500 ccRCC patients ($P < 0.001$, respectively), and HE staining revealed the close relationship between p-mTOR expression and the infiltration of immune cells. The above results also provided rationale for mTOR-targeted therapy of RCC, especially metastatic RCC. To our knowledge, this is the first report that evaluates the prognostic role and interaction to immune infiltration of mTOR in ccRCC patients simultaneously, which revealed p-mTOR could be a prognostic factor and mTOR regulate multiple immune features in ccRCC patients.

Extensive literature reported mTOR was implicated in the initiation and progression of RCC (*Darwish et al., 2013*; *Kruck et al., 2010*; *Liontos et al., 2017*; *Rausch et al., 2019*). Kruck compared the expression profiles of *mTOR* and p-*mTOR* (S2448) in ccRCC and adjacent kidney tissues ($n = 10$), which found the ratio of p-mTOR to mTOR was increased in ccRCC (*Kruck et al., 2010*), which was also observed in the present study. A subsequent TMA analysis revealed that elevated p-mTOR, instead of mTOR, was associated with tumor aggressiveness and impaired OS in 342 primary and 90 metastatic ccRCC patients, and univariate survival analysis displayed that elevated p-mTOR was a predictive marker of poor OS (*Rausch et al., 2019*), which validated our present study. Furthermore, Liontos reported that combination of increased p-mTOR and low vascular endothelial growth factor (VEGF) expression was negatively associated with OS in 79 patients with mRCC who were refractory to first-line sunitinib treatment (*Liontos et al., 2017*). Darwish evaluated the prognostic significance of multiple biomarkers in mTOR pathway components, including p-mTOR in 419 patients with ccRCC, and found a negative correlation between the marker score of mTOR pathway components and the recurrence-free survival (RFS) and CSS in nonmetastatic ccRCC patients (*Darwish et al., 2013*). Notably, p-mTOR, but not mTOR, was also associated with aggressive features and poor prognosis in multiple cancers, such as lung, live and esophagus (*Hirashima et al., 2010*; *Lu et al., 2020*; *Su et al., 2016*). These literatures revealed the negative role of p-mTOR in the patient's survival. Through large-scale proteogenomic analysis, several researches demonstrated mTOR alteration or activation in RCC, but most of them were not involved in its prognosis (*Cancer Genome Atlas Research Network, 2013*; *Chen et al., 2016*; *Fan et al., 2020*; *Han et al., 2017*; *Zhang et al., 2017*). Using integrative proteogenomic analysis, Zhang found that p-mTOR expression level was significantly correlated with improved outcomes in 32 major types of cancer including RCC, which was contradictory to the experimental results (*Zhang et al., 2017*). Similarly, we reported the prognosis of PI3K/AKT/mTOR pathway members in ccRCC using public databases analysis, which demonstrated that mTOR mRNA were positively correlated with OS of over 500 ccRCC patients, and p-mTOR (S2448) was also positively correlated with OS in patients with ccRCC ($n = 445$) (*Fan et al., 2020*). But no experimental evidence has

validated the positive prognosis of mTOR in ccRCC patients yet. Collectively, the prognostic role of mTOR in RCC remains controversial. The prognostic inconsistency of mTOR in RCC might be due to RCC specimens (remarkable heterogeneity, varied sample size and divergence of clinicopathological backgrounds), mutation and amplification of PI3K/AKT/mTOR pathway members, or overexpression of the components of mTORC1 and mTORC2 (*Fan et al., 2020*; *Hua et al., 2019*).

Human immune system could recruit and activate immune guardian-T cells to identify and eliminate malignant cells through antigen-antibody response and cell-mediated immunotoxicity. To this end, tumor immune microenvironment plays a determinative role in tumor survival and progression through mediating the anti-tumor immune response (*Posadas, Limvorasak & Figlin, 2017*; *Saxton & Sabatini, 2017*). Comprehensive investigation of tumor and immune infiltrates would assist to elucidate cancer pathogenesis and develop novel immunotherapy strategies (*Ru et al., 2019*). ccRCC is recognized as one of the more responsive tumors to immunotherapy, and mTOR has been reported to be participate in tumor immunity (*Posadas, Limvorasak & Figlin, 2017*; *Saxton & Sabatini, 2017*). Recent studies reported that mTOR could regulate tumor immunity through modulating the interactions between the stroma and the tumor, thus possibly promote carcinogenesis (*Guri, Nordmann & Roszik, 2018*; *Irelli et al., 2019*). mTOR not only regulated the innate and adaptive immune response through modulating the effector response of innate immune cells such as macrophage, DCs, neutrophils and NKs, but also played a prerequisite role in the development of adaptive immune cells, such as CD4+ T, CD8+ T, Tregs and B cells (*Nazari et al., 2021*). The underlying mechanism behind the association of mTOR expression and immune cells has not been clearly illustrated. Briefly, mTOR could regulate the expression of cytokines/chemokines, including interleukin-10 (IL-10) and transforming growth factor-β (TGF-β), and/or membrane receptors, such as cytotoxic T-Lymphocyte protein 4 (CTLA-4), PD-L1 and PD-1, to modulate tumor immune cells (*Irelli et al., 2019*). In addition, Donnelly demonstrated that mTOR activity was prerequisite for the production of interferon-γ (IFN-γ), a key NK cell effector molecule, and IFN-γ subsequently activated NK cell to regulate immune responses (*Donnelly et al., 2014*). Moreover, Sordi reported inhibition of mTORC1 by rapamycin could increase the migration of DCs to lymph nodes *in vivo* through upregulating CC-chemokine receptor 7 (CCR7) expression (*Sordi et al., 2006*). Immune checkpoint inhibitor represented by PD-1 and PD-L1 monoclonal antibodies has been recommended for mRCC patients. PD-1 (also called PDCD1, CD279) is expressed on immune cells, especially tumor specific T cells, while PD-1 ligand PD-L1 (also called B7-H1, CD274) is expressed on tumor cells as an "adaptive immune mechanism" to escape anti-tumor response (*Han, Liu & Li, 2020*). In the current study, we investigated the relationship between mTOR and immune infiltration in ccRCC. Our data demonstrated mTOR was not only significantly associated with infiltrating levels of numerous immune cell subtypes such as NK and Treg cells, but also positively correlated with various immunoinhibitors such as CD274 (PD-L1) in ccRCC (*Saxton & Sabatini, 2017*). What's more, we also detected rich immune infiltration in ccRCC tissues using HE staining (*Diaz-Montero, Rini & Finke, 2020*), especially in the tissues with higher p-mTOR expression, which

demonstrated that p-mTOR expression was closely associated with intratumoral immune infiltration. These results indicated the underlying mechanism of mTOR in ccRCC development might be involved in tumor immunity, such as immune cells and immunomodulator PD-L1.

In addition to intratumoral immune cells, recent studies began to uncovered the role of inflammatory-related parameters in peripheral blood, such as neutrophil and lymphocyte, during RCC development. There were increasing evidences that inflammatory-related parameters were of prognostic role in mRCC patients. Neutrophilia and thrombocytosis are markers of systemic inflammation, while lymphopenia is related to dysfunctions of the immune system. Kucharz evaluated inflammatory parameters, such as neutrophil, lymphocyte and platelet in 131 mRCC patients treated with Sunitinib, and found that a neutrophil-to-lymphocyte ratio (NLR) lower than 1.64 predicted better outcomes (PFS, OS) (*Kucharz et al., 2019*). Similarly, Tjokrowidjaja reported enhanced NLR was associated with shorter survival adjusting for Memorial Sloan Kettering Cancer Center (MSKCC) variables in 1102 mRCC patients (*Tjokrowidjaja et al., 2020*). To the best of our knowledge, there has been no report about the relationship between mTOR and blood inflammatory cells in RCC yet, and further studies are needed to clarify this issue.

There are some limitations of this study. The first one is the limited sample size for prognostic analysis, and more specimens or even multiple-center clinical studies should be needed for further prospective validation. The second limitation is that only primary ccRCC tissues were enrolled in the present study, and metastatic lesions, such as adrenal gland, lymph node and lung, of mRCC patients, are not included. The prognostic significance and tumor-immune interaction between primary and metastatic lesions of the same patients should be analyzed. Finally, it should be marked that the expression patterns of adjacent renal tissues from RCC patients might differ from benign renal tissue from healthy patients.

## CONCLUSIONS

In summary, the present study detected the expression of mTOR and p-mTOR in ccRCC patients using WB analysis, investigated their prognosis using IHC assay, and explored the interactions between mTOR and immune infiltrates using TIMER and TISIDB databases, which aimed to elucidate the clinical significance of mTOR in ccRCC. We found the ratio of p-mTOR to mTOR was higher in ccRCC than adjacent tissues, p-mTOR was identified as a poor prognostic factor of CSS, and mTOR was associated with numerous immune features in ccRCC patients, which validated mTOR could be implicated in the initiation and progression of ccRCC.

### Funding

This work was supported by Shandong Medical and Health Science and Technology Development Project (2016WS0481), Shandong Provincial Key Research and Development Project (2019GSF107058), and Cultivation Foundation of The First

Affiliated Hospital of Shandong First Medical University (QYPY2019NFSC0601).
The funders had no role in study design, data collection and analysis, decision to publish, or preparation of the manuscript.

## Grant Disclosures

The following grant information was disclosed by the authors:
Shandong Medical and Health Science and Technology Development Project: 2016WS0481.
Shandong Provincial Key Research and Development Project: 2019GSF107058.
Cultivation Foundation of The First Affiliated Hospital of Shandong First Medical University: QYPY2019NFSC0601.

## Competing Interests

The authors declare that they have no competing interests.

## Author Contributions

- Na Li performed the experiments, prepared figures and/or tables, and approved the final draft.
- Jie Chen performed the experiments, prepared figures and/or tables, and approved the final draft.
- Qiang Liu analyzed the data, prepared figures and/or tables, authored or reviewed drafts of the paper, and approved the final draft.
- Hongyi Qu analyzed the data, authored or reviewed drafts of the paper, and approved the final draft.
- Xiaoqing Yang performed the experiments, prepared figures and/or tables, and approved the final draft.
- Peng Gao analyzed the data, authored or reviewed drafts of the paper, and approved the final draft.
- Yao Wang analyzed the data, authored or reviewed drafts of the paper, and approved the final draft.
- Huayu Gao performed the experiments, authored or reviewed drafts of the paper, and approved the final draft.
- Hong Wang conceived and designed the experiments, prepared figures and/or tables, and approved the final draft.
- Zuohui Zhao conceived and designed the experiments, prepared figures and/or tables, authored or reviewed drafts of the paper, and approved the final draft.

## Human Ethics

The following information was supplied relating to ethical approvals (*i.e.*, approving body and any reference numbers):

The First Affiliated Hospital of Shandong First Medical University approved this research (2017-S007).

## Data Availability

The raw measurements are available in the Supplemental Files.

## Supplemental Information

Supplemental information for this article can be found online at http://dx.doi.org/10.7717/peerj.11901#supplemental-information.

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
