# Peer review of "Prognostic significance and tumor-immune infiltration of mTOR in clear cell renal cell carcinoma"

_PeerJ, doi:10.7717/peerj.11901_

## Round 0.1 · original submission · Major Revisions

Consider that reviewer has 2 attached an annotated manuscript to this review.

Reviewer 1 ·

Basic reporting

This paper discussed the expression and prognosis of mTOR and phosphorylated mTOR (p-mTOR) in clear cell RCC (ccRCC) patients, and explored the interactions between mTOR and immune infiltrates in ccRCC. This paper is reading friendly. The overall structure and results are clear.

Experimental design

The design of this experiment is simple and clear. After collecting the data, the author performs correlation analysis and survival analysis by using two existed databases including TIMER and TISIDB to show the relationship between mTOR, p-mTOR and immune infiltrates.

Validity of the findings

1) The correlation between mTOR expression and tumor infiltrating immune cells is too general which just contains the correlation plot. It would be helpful if the author could find other validations through literatures. The mechanism behind the association between mTOR expression and immune cells is still unclear. For example, the author just mentions the statistical results of the correlation in the section of TIMER and TISIDB analyses revealed the relationship between mTOR and immune infiltrates in ccRCC. I think that more biological evidence is needed to validate the statistical results.
2) The survival plot shows that the p-mTOR was identified as an independent predictor of poor CSS with significant p-value (0.002). The author also shows that the p-mTOR is significantly correlated with metastasis, tumor size, and stage. However, the sample size for this plot is pretty small, which is around 150. It’s doubtful whether the conclusion is generalizable to other datasets.
3) As mentioned by the author, controversial conclusions were made by different studies. The role of p-mTOR on the survival is totally reversal. This means that more evidence is needed to validate paper’s conclusion. For example, is there any group showing the negative p-mTOR role on the patient’s survival? Besides, is the conclusion of mTOR consistent with other studies? For example, is there any study showing the significant relationship between mTOR and patient’s survival?

Additional comments

None

Reviewer 2 ·

Basic reporting

The manuscript is well written, authors have shared all the relevant data associated with the experiments. There is need to clarify the figure 5 and 6 and need to better address them in the reporting and discussion.

Experimental design

The research is thought-out and would add more information to the field of Renal cell carcinoma. There are some gaps in the studies that need to be filled

Validity of the findings

The authors have validated their data in human samples and have utilized the available data to validate their findings using a larger data set. However, the authors need to validate some of it in their 145 ccRCC patients as well as in the 3 patients that they chose to show mTOR staining

Additional comments

The authors have a well written and thoughtout manuscript. They have used a small data set to generate the hypothesis and validated it on a larger model.

Annotated reviews are not available for download in order to protect the identity of reviewers who chose to remain anonymous.

---

## Round 0.2 · accepted · Accept

The revision has been properly performed.